# Preliminary Results of Surgical Treatment for Enchondroma Using a Novel Bioactive and Osseoconductive HAP/β–Glucan Bone Substitute FlexiOss^®^—Case Series

**DOI:** 10.3390/jcm14113738

**Published:** 2025-05-27

**Authors:** Daniel Kotrych, Dawid Ciechanowicz, Filip Bielewicz, Andrzej Baryluk, Sebastian Podsiadło, Paweł Ziętek

**Affiliations:** 1Department of Children Orthopaedics and Musculoskeletal Oncology, Pomeranian Medical University, 71-281 Szczecin, Poland; 2Department of Orthopaedics, Traumatology and Musculoskeletal Oncology (DOTMO), Pomeranian Medical University, 71-281 Szczecin, Poland; 3Department of Orthopaedics and Traumatology, Hospital No. 1 in Gliwice, 44-100 Gliwice, Poland

**Keywords:** enchondroma, bone substitute, bone tumor, HAP/β–glucan, surgical treatment

## Abstract

**Background/Objectives**: In the surgical treatment of benign bone tumors, bone substitutes are widely used. However, each of them has its advantages and disadvantages. We decided to study the novel bioactive and osseoconductive HAP/β–glucan bone substitute. **Methods**: We qualified eight patients with enchondroma of the lower limbs for this study, who underwent marginal resection of the lesion with the use of a bone substitute—FlexiOss^®^. During the 12-month follow-up, a series of X-rays and bone scintigraphy were performed. Bone remodeling was assessed using the modified Neer scale (MNS), while functional results were assessed using the MSTS scale. **Results**: Very good functional results were noted in all patients—MSTS = 27–30. In the MNS, Score I was recorded in six cases, and Score II was recorded in two cases. Among the complications in the two patients, a clear serous discharge from the wound was observed. **Conclusions**: The use of the new HAP/β–glucan composite in the treatment of enchondroma initially shows good treatment results.

## 1. Introduction

Benign bone tumors are a heterogeneous group of neoplasms. It is estimated that approximately 40% of all diagnosed bone tumors are benign, with no potential to metastasize. However, this disease may cause pain, swelling, reduced limb function, and even lead to a pathological fracture [1]. Therefore, some tumors qualify for surgical treatment. The method of choice is to remove the lesion (curettage) and to fill the bone defect [2,3]. For this purpose, a bone graft—allogeneic or autologous—or a synthetic bone substitute can be used. Bone autografts show very good clinical results, but this method has some limitations, such as possible complications at the donor site, the limited availability of grafts of appropriate size, and additional procedures that prolong the surgery [4]. The problem with allografts, however, is their low availability and relatively high rate of complications [5,6]. Therefore, in recent years, there has been an increased demand for bone biomaterials, which are an alternative to traditional auto- and allografts. In the United States, more than 2 million bone graft surgeries are performed each year, exceeding the bone biomaterials market value of USD 39 billion in 2013 [7]. One of the most commonly used bone substitute materials in the world is bone cement, which is based on polymethyl methacrylate (PMMA). This method provides good stability and tissue compatibility, but PMMA is not biodegradable and can lead to bone necrosis due to its exothermic reaction [8,9]. Other bone substitutes are synthetic ceramic materials, but they also have a high rate of complications [10,11]. Among the available bone substitutes, flexible hydroxyapatite-based composites are becoming increasingly popular. These biomaterials are composed of hydroxyapatite (HAP) granules and polysaccharide polymer (β-1,3-glucan). It is characterized by high bioavailability and bioabsorption, thanks to the use of curdlan as an ingredient [12]. Curdlan exhibits significant water absorption capacity and, therefore, creates optimal conditions for the circulation of oxygen, nutrients, and metabolic waste, which enables the regeneration and growth of the tissues surrounding the biomaterial. The basis of the preparation is hydroxyapatite and curdlan. Hydroxyapatite (HAP-Ca_10_(PO_4_)6(OH)_2_) is a mineral scaffold responsible for the mechanical strength of bones, which is why it is widely used as an artificial bone substitute in the form of granules and porous scaffolds. It is valued for its natural structure, bioactivity, biocompatibility, ability to support bone tissue growth (osteoconductivity), and its lack of immunogenicity and toxicity. Due to its dielectricity (the poor conduction of electrical currents), it can be used in implants in patients requiring physical therapy treatments [13]. Its disadvantages include fragility, low resistance to fracture, poor resorption, and low surgical convenience (with no possibility of adjusting the shape). In order to improve mechanical properties and convenience, additional bonding and elasticity-increasing substances were used [14]. Curdlan is a water-insoluble linear beta-1,3-glucan, a high-molecular-weight glucose polymer, and a sugar polymer with a triple helix structure. It is produced by the Gram-negative bacterium Alcaligenes faecalis var. myxogenes, although some bacteria of the Agrobacterium genus also have this ability [15]. The triple helix of curdlan allows hydroxyapatite granules to penetrate its structure, which leads to the formation of a hydroxyapatite–polymer biocomposite. Thanks to this structure, this material easily absorbs various fluids, such as blood, physiological fluids, or drug solutions, which make it flexible and elastic. The HAP/β–glucan composite creates a scaffold for osteogenic cells and stem cells due to its high porosity [16]. Moreover, it is non-toxic to the surrounding tissues and to the bone tissue itself. It is also biocompatible and bioactive in the sense that it attracts calcium and phosphate ions to the implant area, leading to the deposition of natural hydroxyapatite in the tissue. The biocomposite also promotes the growth of its own bone tissue by allowing the settlement of bone-forming cells. All these features accelerate the regeneration process of the damaged area without causing inflammation. Hydroxyapatite–polymer biocomposites are mainly used in the treatment of bone defects with low or medium mechanical loads, such as inflammatory defects resulting from metabolic disorders or trauma [12]. Up until now, there have been no reports investigating the use of the HAP/β–glucan composite in the treatment of benign bone tumors. Therefore, the aim of this publication is to present the results of the treatment of eight patients with enchondroma in the lower limbs.

## 2. Materials and Methods

This is a retrospective case series involving data collection from patients treated in our Institution due to benign bone tumors with the use of hydroxyapatite/glucan composites (FlexiOss^®^, Medical Inventi S.A., Lublin, Poland). The substitute has EC Certificate No. 1434-MDD-269/2021 and has been approved for use in humans in the European Union. All patients were treated in 2023. Patients were qualified for surgical treatment according to the 2024 BOOM consensus from Birmingham [17]. The diagnosis of a benign bone tumor was made on the basis of radiological examinations. The study included patients whose X-ray, MRI, and bone scintigraphy clearly indicated the benign nature of the lesion. All results were assessed individually by an orthopedist and radiologist with experience in diagnosing and treating bone tumors. The inclusion criteria were the following: (1) >18 years old; (2) diagnosis of a benign bone tumor or enchondroma (3) indications for surgical treatment and resection of the lesion (pain complaints); (4) location of the lesion in the lower limb; and (5) completion of all follow-up visits (2 weeks, 6 weeks, 6 months, and 12 months after surgery). This study included a total of eight patients. The following data were collected: age, gender, symptoms, the location and size of the bone lesion (largest dimension), the amount of bone substitute used during surgery, postoperative histopathological diagnosis, results on the VAS (pre- and postoperative at follow-up visits) and MSTS functional scale (in last follow-up visit), and reported complaints during the follow-up period. The research related to human use complied with all the relevant national regulations and institutional policies set forth by the tenets of the Helsinki Declaration.

In all patients, marginal resection of the benign bone tumor was performed using a bone curette and bone cutter. Before the procedure began, according to the manufacturer’s recommendations, the bone substitute was soaked in sterile saline for 15 min. After that time, the bone substitute was placed (Figure 1). The entire procedure was performed under fluoroscopic control. On the first day after surgery, all patients were in a supine position with the operated limb elevated and cold compresses applied to the operated area. From the 2nd day, walking on 2 elbow crutches was allowed, and isometric muscle exercises were implemented. With each successive day of rehabilitation, the load on the operated limb during walking was increased. Full weight bearing of the operated limb was implemented 4 weeks after surgery. All patients had 5 radiographs taken—before the surgery and at 0, 2, 6, and 12 months after the surgery. Moreover, during the last follow-up visit (12 months after surgery), bone scintigraphy in the SPECT-CT module was performed to assess the metabolism of bone tissue. Radiographs 12 months after surgery were analyzed using the modified Neer classification by two orthopedic specialists experienced in the treatment of benign bone lesions [18]. This scale is used to determine what kind of treatment, if any, is necessary after the curettage of a benign bone tumor. Scores from I to IV are assigned based on the size of the radiolucent area in the image. The percentage of bone diameter visible as radiolucent correlates with the danger of the fracture. Scores I and II are described as “healed” and “healed with defect”, respectively (as radiolucent areas take up less than 50% of the diameter of the bone), and no additional treatment is necessary. Score III patients should continue restricting the activity of the affected limb, and reoperation should be considered. Score IV denotes the need for reoperation.

## 3. Results

In all patients, the postoperative histopathological result indicated a benign bone lesion or enchondroma without atypia (Table 1). Among the symptoms experienced, all patients reported pain in the area of the bone tumor. In three cases, there was also palpable edema. In all patients, no perioperative complications were noted after surgical treatment. All patients spent 3–4 days in the hospital, the first day of which was the day of admission; on the second day, the surgery was performed; and on the third or fourth day, the patients were discharged home.

In the modified Neer classification (MNC), Score I was recorded in six cases (75%), and Score II was recorded in two cases (25%). In bone scintigraphy, a moderately increased tracer (radiopharmaceutical) accumulation was noted in all cases, which may suggest regenerative and repair remodeling processes following bone tumor resection [Figure 2]. No infections, bone fractures, or neurological complications were noted during the 12-month follow-up. In six of the eight patients, the maximum score on the MSTS scale (30 points) was noted. In two patients, due to intermediate pain, which periodically hindered daily functioning, a lower score on the MSTS scale was noted (Table 2).

In two patients (25%), at the first follow-up visit (2 weeks after the procedure), a clear serous discharge from the wound was observed. There were no signs of inflammation or infection. At the next follow-up visit (6 weeks after the procedure), a normal local condition and a healed postoperative wound were observed. In six of the eight patients, there was no pain in the operated limb during the follow-up. After resection of the femoral neck tumor, the patient experienced periodic pain, assessed as Scores 3–4 on the VAS (case 1). In one patient, 12 months after the surgical treatment, periodic foot swelling occurred without pain (case 3) (Table 3).

## 4. Discussion

Bone defects caused by tumor resection are a common orthopedic problem. In the case of benign bone tumors, the method of treatment depends on the presented symptoms. In the case of symptomatic tumors, the most common treatment is marginal resection (curettage). To restore the structural integrity of the bone, a substitute must be used. The use of an autogenous bone graft, although considered to be the gold standard, is associated with possible complications at the donor site. Moreover, in the case of older patients, bone tissue may not be suitable for grafting due to osteoporotic changes. They can also be replaced by allografts, which are unfortunately characterized by low availability, possible graft rejection by the recipient, and the possible spread of viral diseases. Therefore, the development of artificial biomaterials suitable for repairing bone defects is an important issue in the field of medicine and materials science. The ideal bone substitute material should be biocompatible with natural bone tissue, support the bone regeneration process, be non-toxic to human tissue, and be comfortable and easy to implant during surgery. One of the new materials that has shown promising results is the bi-phasic ceramics–glucan composite used in the patients described in this article. It exhibits high biocompatibility and induces efficient osseointegration after implantation [13,19]. Moreover, it is characterized by high flexibility, which not only affects its easy implementation possibilities but also allows for very good adjustment to the bone defect. This was confirmed by the study conducted by Borkowski et al., where after the use of the described substitute in the proximal tibial metaphysis of rabbits, precise adjustments to the shape/size of the bone cavity and stable retention at the implantation site were observed. The authors also noted good reorganization of the bone tissue around the implantation site and the formation of plexiform bone at the bone–implant interface one month after the procedure, without signs of inflammation or features of graft rejection. This suggests that the bi-phasic ceramics–glucan composite is characterized by high biocompatibility and osseointegration [20]. In another study, Borkowski et al. presented a case report detailing the use of the HAP/β–glucan composite in a femoral defect. Twelve months after the procedure, significant mineralization of the implant, penetration of newly formed bone tissue into the three-dimensional structure of the composite, and bone tissue reorganization were observed. Nineteen months after the procedure, high mineralization and complete bone tissue regeneration were observed at the implant site. In the second patient, a smaller composite element was implanted into the tibial defect. After three months, both the slight mineralization of the implanted composite and radiological signs of bone union were observed. Twelve months after the procedure, high radiolucency and complete bone tissue synostosis were demonstrated in the implant area [12]. Similar effects were observed in our study, where the radiological assessment of the patients according to the Neer scale was performed where Score I was defined as healed bone in six patients, and in two cases, Score II was defined as healed bone with a defect (Figure 3).

In the cases presented in this manuscript, good postoperative results were noted. Although all patients underwent open surgery, very good functional results were observed (MSTS = 27–30 points), and pain relief was achieved in five out of six cases. In the case of early unimprovement, the symptoms may have resulted from the primary location of the bone tumor—the femoral neck—which determined the need for a larger surgical approach and interference in the part of the bone that was exposed to high mechanical loads [Figure 4]. In this patient, worse bone remodeling was also observed in the X-ray compared to the two other cases. As shown in the study by Yong-Cheng Hu et al., in the case of bone tumors located in the femoral neck, pain that initially occurs and limping may subside over a longer follow-up period [21]. Functional results after curettage in benign bone tumors are rated as very good and comparable with all the most commonly used bone substitutes—PMMA, allografts, autografts, or Bi-Phasic Bone Substitute—CERAMENT [10,22,23,24]. In two patients, increased serous discharge from the wound was observed within 2 weeks of surgical treatment. The literature describes a similar problem in cases where the Bi-Phasic Bone Substitute is used. This is confirmed in the study conducted by Zrodowski et al., where, in a group of 38 patients, the above complications were visible in 10.5% of cases [11]. Another study indicated that calcium sulfate carries a cytotoxic effect—dissolution—which leads to an acidic microenvironment that is responsible for local inflammatory processes at the site of implantation in human bones [25]. However, studies evaluating the clinical effect of the Bi-Phasic Bone Substitute indicate a lack of cytotoxicity, and the increased serous discharge from the wound could be caused by the development of the new calcium material [26,27]. Also, in cases using the HAP/β–glucan composite, the lack of toxicity to osteoblasts and the surrounding tissues was demonstrated [14,20]. In addition, Borkowski et al. indicated that, in the case of bone substitutes that increase in volume after contact with fluids, excessive swelling may occur, which may lead to delayed wound healing. They showed that local inflammation leads to a decrease in plasma pH, which intensifies the above effect. In the case of neutral plasma pH, this effect was not observed [12]. Therefore, it seems reasonable to limit the body’s inflammatory response during the postoperative wound-healing period in this group of patients. One of the limitations of our study is the small group of patients. The study, however, is intended to show preliminary results after the use of the new HAP/β–glucan composite in the treatment of benign bone tumors. Our case series shows promising results and provides an introduction to conducting further prospective studies.

## 5. Conclusions

The use of the new HAP/β–glucan composite in the treatment of enchondroma initially shows good treatment results. It can be safely used as a bone substitute in patients, and it provides good functional results and reductions in pain intensity. Based on radiological studies, the HAP/β–glucan composite is characterized by very good biocompatibility and osseointegration. Moreover, due to its flexibility and solid consistency, it is easy to apply and adapts very well to the bone defect. Further study is required to assess the number of complications (especially instances of serous discharge from the wound) experienced after this composite’s use.

## Figures and Tables

**Figure 1 jcm-14-03738-f001:**
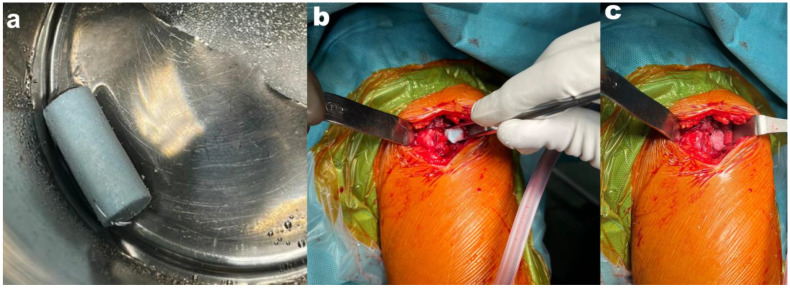
Intraoperative photographs: (**a**) the bone substitute is soaked in sterile saline for 15 min; (**b**,**c**) placement of a bone substitute in the bone bed after resection of the lesion.

**Figure 2 jcm-14-03738-f002:**
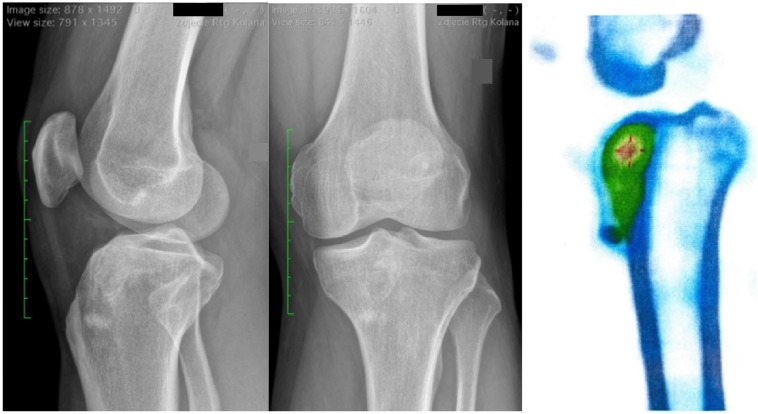
The X-ray views after 6 weeks of follow-up showing proper bone defect healing. Bone SPECT-CT proved physiological bone metabolism within the reconstructed defect.

**Figure 3 jcm-14-03738-f003:**
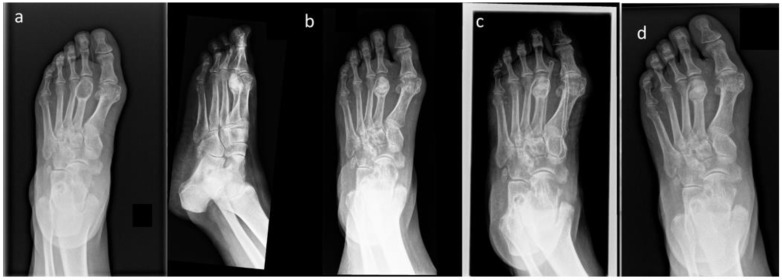
The X-ray views (**a**) before the surgery–benign bone lesion in second metatarsal bone; (**b**) postoperative image after resection of the bone lesion with the use of the bioactive and osseoconductive HAP/β–glucan bone substitute; and (**c**,**d**) the same case after 12 months.

**Figure 4 jcm-14-03738-f004:**
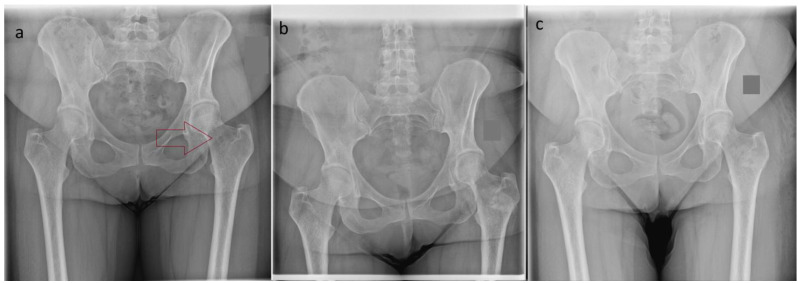
The X-ray views (**a**) before the surgery–benign bone lesion in the left femoral neck (red arrow); (**b**) postoperative image after the resection of the bone lesion with the use of the bioactive and osseoconductive HAP/β–glucan bone substitute; and (**c**) the same case after 12 months.

**Table 1 jcm-14-03738-t001:** Characteristics of all patients.

Case Number	Gender	Age at Surgery (Years)	Tumor Localization	Tumor Size [mm]	Tumor Characteristic
1	F	42	Femur (neck)	11	Enchondorma
2	F	66	Foot (metatarsus)	30	Enchodnorma
3	F	20	Fibula	18	Endchondroma
4	F	48	Ilium	25	Enchindroma
5	M	55	Tibia	33	Enchondroma
6	F	18	Fibula	40	Enchondroma
7	F	42	Fibula	10	Enchondroma
8	F	44	Femur (shaft)	50	Enchondroma

**Table 2 jcm-14-03738-t002:** Detailed results on the MSTS (Musculoskeletal Tumor Society Scoring) functional scale.

Case Number	Pain	Function	Emotional	Supports	Walking	Gait	MSTS Result
1	Intermediate (4)	Intermediate (4)	Intermediate (4)	None (5)	Unlimited (5)	Normal (5)	27
2	No pain (5)	No restriction (5)	Enthused (5)	None (5)	Unlimited (5)	Normal (5)	30
3	No pain (5)	No restriction (5)	Enthused (5)	None (5)	Unlimited (5)	Normal (5)	30
4	No pain (5)	No restriction (5)	Enthused (5)	None (5)	Unlimited (5)	Normal (5)	30
5	No pain (5)	No restriction (5)	Enthused (5)	None (5)	Unlimited (5)	Normal (5)	30
6	No pain (5)	No restriction (5)	Enthused (5)	None (5)	Unlimited (5)	Normal (5)	30
7	Intermediate (4)	Intermediate (4)	Enthused (5)	None (5)	Unlimited (5)	Normal (5)	28
8	No pain (5)	No restriction (5)	Enthused (5)	None (5)	Unlimited (5)	Normal (5)	30

**Table 3 jcm-14-03738-t003:** The table presents the results of the modified Neer classification (MNC) and the assessment of pre- and postoperative pain in patients on the VAS, as well as all complications during the 12-month follow-up.

Case Number	MNC	VAS Score Pre-Operative	VAS Score Postoperative	Complications
1	Score II	6	3–4	Serum leakage from the wound—2 weeks after surgery
2	Score I	5	0	None
3	Score I	4	0–1	Periodic foot swelling
4	Score I	7	4	None
5	Score I	6	3	None
6	Score I	5	2	None
7	Score II	3	2	Serum leakage from the wound—2 weeks after surgery
8	Score I	5	0	None

## Data Availability

The raw data supporting the conclusions of this article will be made available by the authors on request.

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
