# Peer review of "Preliminary Results of Surgical Treatment for Enchondroma Using a Novel Bioactive and Osseoconductive HAP/β–Glucan Bone Substitute FlexiOss^®^—Case Series"

_jcm, 2025, doi:10.3390/jcm14113738_

Round 1

Reviewer 1 Report

Comments and Suggestions for Authors

This manuscript presents a retrospective case series of eight patients with enchondromas in the lower limbs treated with FlexiOss®, a new HAP/β-glucan-based bone substitute. All patients underwent marginal resection of the tumor and bone defect filling. Radiological and functional outcomes were monitored over 12 months. The results showed promising functional scores (MSTS 27–30), low complication rates, and good osseointegration in most cases.

  • Line 20: Typo in “MNC”

  • Line 23: Clarify what “good treatment results” means

  • Line 79: Clarify the novelty: is this truly the first report in benign tumors or only in enchondroma?
  • Lines 118–124: Rephrase
  • First part of discussion is very generic and ripetitive. I suggest a deep revision.

Author Response

We would like to thank the reviewer for all comments on our manuscript. They have been very helpful and we hope that the manuscript will be more interesting to readers after the corrections. Below is our response to each point.

Comment: Line 20: Typo in “MNC”

Respond: The correction was made

Comment: Line 23: Clarify what “good treatment results” means 

Respond:  Due to limitations in the abstract structure, we used the abbreviation 'good treatment results', which refers to the results mentioned above, i.e.: low complication rate, good functional results and acceptable bone remodelling after 12 months.

Comment: Line 79: Clarify the novelty: is this truly the first report in benign tumors or only in enchondroma?

Respond: To the best of our knowledge, this is the first study describing the use of HAP/β-glucan bone substitute FlexiOss® in patients with benign bone tumors.

Reviewer 2 Report

Comments and Suggestions for Authors

The authors of this paper address an important topic concerning the surgical treatment of benign bone tumor enchondroma with bone biomaterial. The methodology lacked information on the pain examination conducted on the VAS scale (in what period after surgery or at each follow-up visit), as well as on the MSTS functional scale (components of the examination, scoring, interpretation). The title of the figure should be below the figure. The results were analyzed according to the methods presented. The tables are well described. The discussion is conducted correctly. The references are typical. Current literature from the last 5 years accounts for 33%. Two items are self-citations. Despite the small research group, the topic may become the basis for further research in other centers.

Author Response

We would like to thank the reviewer for all comments on our manuscript. They have been very helpful and we hope that the manuscript will be more interesting to readers after the corrections. Below is our response to each point.

Comment: The methodology lacked information on the pain examination conducted on the VAS scale (in what period after surgery or at each follow-up visit), as well as on the MSTS functional scale (components of the examination, scoring, interpretation).

Respond: The VAS scale assessment was conducted during a medical interview the day before surgery and after surgery during follow-up visits at the orthopedic clinic. The VAS scale data from the last follow-up visit (12 months after surgery) were included in the publication. MSTS scale results were collected during the last follow-up visit (12 months after surgery). We entered these data into the manuscript.

Comment: The title of the figure should be below the figure.

We changed that in the manuscript.

Reviewer 3 Report

Comments and Suggestions for Authors

The authors have described the application of a bone substitute with a biphasic composition containing both ceramics and glucan for the correction of bone deficits in patients with benign bone tumors, in this case enchondromas.  This a case series report.  The authors should also compare their results with those of other authors, as according to their report in the USA the use of biocompatible materials for the correction of bone deficitis is growing.  Despite the small number of patients this is an extremely interesting paper.

Comments on the Quality of English Language

The use of the English language needs minor improvement.

Author Response

We would like to thank the reviewer for all comments on our manuscript. They have been very helpful and we hope that the manuscript will be more interesting to readers after the corrections.

In the discussion we compare the use of flexioss to the most commonly used bone substitutes in Poland. However, we will add a comparison to other bone substitute materials.